# A Two-Staged Feature Extraction Method Based on Total Variation for Hyperspectral Images

**Chunchao Li, Xuebin Tang, Lulu Shi, Yuanxi Peng and Yuhua Tang \***

State Key Laboratory of High Performance Computing, College of Computer Science and Technology, National University of Defense Technology, Changsha 410073, China; lcc@nudt.edu.cn (C.L.); xbtang@nudt.edu.cn (X.T.); sll@nudt.edu.cn (L.S.); pyx@nudt.edu.cn (Y.P.)
**\*** Correspondence: yhtang@nudt.edu.cn

**Abstract:** Effective feature extraction (FE) has always been the focus of hyperspectral images (HSIs). For aerial remote-sensing HSIs processing and its land cover classification, in this article, an efficient two-staged hyperspectral FE method based on total variation (TV) is proposed. In the first stage, the average fusion method was used to reduce the spectral dimension. Then, the anisotropic TV model with different regularization parameters was utilized to obtain featured blocks of different smoothness, each containing multi-scale structure information, and we stacked them as the next stage's input. In the second stage, equipped with singular value transformation to reduce the dimension again, we followed an isotropic TV model based on split Bregman algorithm for further detail smoothing. Finally, the feature-extracted block was fed to the support vector machine for classification experiments. The results, with three hyperspectral datasets, demonstrate that our proposed method can competitively outperform state-of-the-art methods in terms of its classification accuracy and computing time. Also, our proposed method delivers robustness and stability by comprehensive parameter analysis.

**Keywords:** feature extraction; hyperspectral image; total variation; smoothing

## 1. Introduction

Hyperspectral imaging technology is based on multi-spectral imaging, in the spectral range from ultraviolet to near-infrared, using an imaging spectrometer to continuously scan within tens or hundreds of spectral bands of the scenes. Therefore, hyperspectral images (HSIs) not only capture spatial features but also obtains rich spectral information from each pixel, which can achieve the classification and recognition of the target objects more efficiently than traditional images. Nowadays, many HSIs passively acquired on satellite or airborne have broad ranges of land cover; they are widely used in many fields such as urban mapping [1], agriculture [2], forest [3], and environmental monitoring [4]. In addition, HSIs can also be obtained by active remote sensing technology [5], which usually utilizes wide spectral light sources [6] to replace the sun to illuminate the scenes and which play a significant role in object detection [7] and recognition [8]. HSI classification has always been a hot topic of application among these fields. It can provide high-level intuitive judgment and interpretation, especially for land use and analysis.

However, some characteristics of HSIs bring difficulties to its application. Firstly, a few pixels of HSI may represent the land cover of tens of square meters, resulting in some specific samples availability being limited, which will lead to low accuracy when directly using the spatial information of HSIs for scene detection. In addition, as the number of bands rises, the amount of data expands sharply, and adjacent bands are highly correlated, noise and redundant information are relatively increased, especially for active technology because of the stability of the light source, which will also affect the classification accuracy. More importantly, these characteristics will bring considerable time and storage overhead in the computing of related algorithms.

Therefore, effective and discriminative feature extraction (FE) processing has become the key to hyperspectral technology [9], and many methods have been applied in the hyperspectral community. Regarding HSIs as three-dimensional cube data, data projection and transformation are commonly used FE methods, such as principal components analysis (PCA) [10], independent component analysis (ICA) [11], and maximum noise fraction (MNF) [12], which are dedicated to linearly transforming the data into a low-dimensional feature space and which reduce the band dimension of HSIs. In addition, supervised linear transformation methods are used, such as linear discriminant analysis (LDA) [13], and some extended versions, including low-rank discriminant analysis (LRDA) [14], locally weighted discriminant analysis (LWDA) [15], and flexible Gabor-based superpixel-level unsupervised linear discriminant analysis [16]. Considering the nonlinear characteristics of HSIs, some techniques using kernel methods have been proposed, such as kernel PCA (KPCA) [17], which can obtain a linearly separable training set by nonlinearly mapping data to a high-dimensional space. What's more, manifold learning methods [18] have been continuously developed, and some advanced methods include GPU parallel implementation of isometric mapping [19], which can greatly accelerate the speed of data transformation. Spatial–spectral multiple manifold discriminant analysis (SSMMDA) [20] can explore spatial–spectral combined information and reveal the intrinsic multi-manifold structure in HSIs. In [21], a novel local constrained collaborative representation model was designed; it can characterize the collaborative relationship between sample pairs and explore the intrinsic geometric structure in HSI. Projection and transformation techniques usually look for the separability of the data in the transformed feature space. Generally, only the internal pattern of the data itself is considered, and the specific physical characteristics are usually lost, especially the continuous correlation of the spectral information in HSI. In addition, data mapping methods are often used as one of the steps of FE, such as completing dimension reduction.

When focusing on the continuous and high correlation of bands, some methods based on band selection and clustering [22] have been proposed and used to mitigate the "curse of dimensionality" problem. The combination of image fusion and recursive filters was applied in [23]. In [24], Wang et al. designed a method of band selection based on optimal neighborhood reconstruction, which exploited a recurrence relation for the optimal solution. Tang et al. [25] proposed a hyperspectral band selection method based on spatial–spectral weighted region-wise multiple graph fusion-based spectral clustering. In [26], an unsupervised BS approach based on an improved genetic algorithm was proposed, which utilized modified genetic operations to reduce the redundancy of selected bands. Band selection methods fully excavate the spectral characteristics of HSI, but it often combines spatial information predominately for the FE process in practical applications.

The method of using energy functional optimization, which is a common technique of traditional image processing, has been widely developed in hyperspectral FE. It's aim is to minimize a constrained loss function, which usually contains fidelity and regularization terms, and play an important role in restoring and noise reduction for HSI [27]. In [28], an orthogonal total variation component analysis (OTVCA) was proposed and used a cyclic descent algorithm for solving the minimization problem. Duan et al. [29] showed a fusion and optimization framework of dual spatial information for more feature exploration. A novel multidirectional low-rank modeling and spatial–spectral total variation (MLR-SSTV) model was proposed in [30] for removing HSI mixed noise, and they developed an efficient algorithm for solving the derived optimization based on the alternating direction method of multipliers. By adding different regularization term constraints, a $l_0$-$l_1$ hybrid total variation ($l_0$-$l_1$HTV) was presented in [31], which yielded sharper edge preservation and obtained superior performances in HSI restoration. But they usually have more computational overhead because numerous iterations are needed to solve the optimization problem.

In addition, with the development of deep learning, many deep FE methods have been proposed for image processing. In [32], Zhu et al. proposed a defogging network based on dual self-attention boost residual octave convolution, which effectively enhances

the definition of foggy remote sensing images. As for HSIs, rich spectral information needs to be considered, and convolutional neural networks are also widely used to extract the spectral and spatial information of HSIs [33,34]. Li et al. [35] proposed a double-branch dual-attention mechanism network (DBDA) for HSI classification, where the two branches enabled to refine and optimize the extracted feature maps. Additionally, some remarkable models, such as ResNet [36] and DenseNet [37], have also been continuously developed. The method combining the scattering transform with the attention-based ResNet was given in [38], which provided higher unmixing accuracy when using limited training data. In [39], Xie et al. proposed the MS-DenseNet framework, which introduced several dense blocks to fuse the multiscale information among different layers for the final HSIs classification. To achieve effective deep fusion features, a pixel frequency spectrum feature based on fast Fourier transformation was presented in [40] and introduced a 3D CNN, which showed that adding the presented frequency spectrum feature into CNNs can achieve better recognition results. However, deep learning techniques often need sufficient training sets to achieve their excellent performance [9], but the samples are limited in HSI. Similarly, they often need a lot of computing resources and expenses.

For aerial remote sensing HSIs processing and their land cover classification, there are several core problems in the FE method to be considered and addressed. According to the different sensors and their scanning positions, the spatial resolution of different HSIs is diverse. Developing the FE method suitable for multi-scale HSIs is the key. At the same time, FE need to fully excavate the spatial characteristics of HSI so that the features information of various classes can be fully retained, especially considering the limited samples of individual classes. On the other hand, computing overhead has increasingly become the bottleneck of tasks, and it is necessary to design lightweight methods for real-time applications. Stemming from a motivation to solve these key issues, in this article, an efficient two-staged hyperspectral FE method based on total variation (TV) [41] is proposed. In the first stage, in order to extract multi-scale structural information, we used the anisotropic total variation model (ATVM) [42] under numerical solution to process the average fused data. The model, with different regularization parameter outputs, featured blocks of different smoothness, each containing multi-scale structure information and then stacking them as the next stage's input. In the second stage, in order to prevent weakening of the feature representation of small sample classes and to fully strengthen the information of all classes, we used an isotropic total variation model (ITVM) based on the split Bregman algorithm [43] to process the data after SVD of the stacked block, where complete dimension reduction and global smoothing was performed. Finally, it was fed to the spectral classifier for the classification task. The main contributions of our study are summarized as follows:

1.  This work innovatively proposes an efficient two-staged FE method based on total variation for HSI. Based on different solutions of anisotropic and isotropic models, it successively completes the extraction of multi-scale structure information and detail smoothing enhancement of HSI, improving the discriminative ability of different land covers.
2.  Our design has no complex framework or redundant loop, which greatly reduces computational overhead. When compared with many state-of-the-art algorithms, our method can significantly outperform in classification performance and computing time, especially achieving better results in most classes.
3.  We give a sufficiently detailed parameter analysis and give the reasonable value and change explanation of each parameter. There is no need to reset parameters for diverse datasets, and the results show that our method has strong robustness and stability, strengthening the advantages in hyperspectral practical application.

The remainder of this article is structured as follows. In Section 2, the proposed method is described. Section 3 shows experimental results and comparisons using three real datasets. Section 4 gives parameter analysis and discussion in detail. The conclusions and future work are presented in Section 5.

## 2. Proposed Method

This section will be divided into three parts. First, the ATVM under the numerical solution of the first stage is described. The second part gives ITVM based on split Bregman algorithm, and the third part shows the overall design of the proposed method.

### 2.1. ATVM

The energy function optimization model often adds different regularization terms, such as TV terms, to maintain the smoothness and suppress the noise of the image. The ATVM under numerical solution, which is derived from relative TV [44], can effectively decompose the structure and texture in an image and strengthen the retention of structural information. The general form aims to solve the following optimization problem:

$$argmin_F \sum_p \left\{ (F_p - R_p)^2 + |(\nabla F)_p| \right\}. \tag{1}$$

$R$ denotes a single-band image of the input raw HSI, $F$ is the output featured image of the corresponding band, and $p$ denotes the index of the two-dimensional image pixels. The TV term can be written in the following anisotropic form:

$$\sum_p |(\nabla F)_p| = \sum |(\nabla_x F)_p| + |(\nabla_y F)_p|. \tag{2}$$

To better extract the structure, the improved model is as follows,

$$argmin_F \sum_p \left\{ (F_p - R_p)^2 + \lambda \cdot \left( \frac{D_x(p)}{L_x(p) + \varepsilon} + \frac{D_y(p)}{L_y(p) + \varepsilon} \right) \right\}, \tag{3}$$

where $\lambda$ is the regularization parameter, and $\varepsilon$ is a small positive number to prevent the divisor from being 0, $D_x(p)$ and $L_x(p)$ are, respectively, called windowed total variations and windowed inherent variation in the x direction for pixel $p$, expressed as

$$D_x(p) = \sum_{q \in R_p} g_{p,q} \cdot |(\nabla_x F)_q|,$$
$$L_x(p) = |\sum_{q \in R_p} g_{p,q} \cdot (\nabla_x F)_q|. \tag{4}$$

where $D_y(p)$ and $L_y(p)$ represent the $y$ direction, the definition and calculation are the same as $x$ direction. In Formula (4), $q$ is the index of all pixels in a square window centered on $p$, and $g$ is the weighting function.

$$g_{p,q} \propto exp(-\frac{(x_p - x_q)^2 + (y_p - y_q)^2}{2\sigma^2}), \tag{5}$$

where $\sigma$ specifies the scale size of filtering elements of the window. Then the regularization term in Formula (3) can be approximately calculated as follows:

$$\sum_p \frac{D_x(p)}{L_x(p) + \varepsilon} = \sum_q u_{x\,q} w_{x\,q} (\nabla_x F)_q^2,$$
$$u_{x\,q} = \left( G_\sigma * \frac{1}{|G_\sigma * \nabla_y F| + \varepsilon} \right)_q,$$
$$w_{x\,q} = \frac{1}{|(\nabla_x F)_q| + \varepsilon_2}, \tag{6}$$

where $G$ is the Gaussian filter with standard deviation $\sigma$, $*$ is the convolution symbol, and $\varepsilon_2$ is a small value to prevent division by zero. The calculation in the $y$ direction is the same way as stated above.

Then, with these operators, Formula (3) can be written in the following matrix form:

$$(v_F - v_R)^T (v_F - v_R) + \lambda \left( v_F^T C_x^T U_x W_x C_x v_F + v_F^T C_y^T U_y W_y C_y v_F \right). \tag{7}$$

Taking the derivative of Equation in (7), one can obtain the following linear equation:

$$v_F = (1 + \lambda M)^{-1} v_R,$$
$$M = C_x^T U_x W_x C_x + C_y^T U_y W_y C_y, \tag{8}$$

where $v_F$ and $v_R$ represent the column vectors of $F$ and $R$, respectively, $C_x$ and $C_y$ are forward difference operators, and $U$ and $W$ are diagonal matrices, where the value comes from $u$ and $w$ in their respective directions.

The numerical solution of ATVM is briefly given in analytical Formula (8), which mainly includes two steps: calculation of coefficient matrix $M$ and solution of linear equation. The two key parameters, $\lambda$, can control the smoothness, and $\sigma$ specifies the scale size. Here, we creatively used the scale size parameter $\sigma$ as the loop condition, entering the fixed $\sigma$ as the maximum, and extracting structural information at multiple scales with a small number of loops. In other words, we use few loops to extract the features of different scales in one output, without setting the number of iterations in advance, reducing redundant iterations. The calculation process of the whole model is briefly summarized in Algorithm 1. In line 5–7, we use the same coefficients to solve the linear equation for each dimension of the image.

---

**Algorithm 1:** The ATVM under Numerical Solution to Extract Different Scale Structure

---

1: **Input:** input raw image $R^n$; regularization parameter $\lambda$; scale size parameter $\sigma$;
2: **Initialize:** $T = R$; setting $\varepsilon = 0.001$ and $\varepsilon_2 = 0.01$ for enough small; $\sigma_t = \sigma$;
3: While $\sigma_t \geq 0.5$
4:    $M = computeCoefficient(T, \sigma_t)$;
5:     for $i = 1 : n$
6:      $T^i = solveLinearEquation(R_i^n, M, \lambda)$; $R_i^n$ denotes the $i - th$ dimension of $R^n$.
7:     end;
8:      $T = [T^1, T^2, \cdots, T^n]$,
9:   $\sigma_t = \sigma_t / 2.0$;
10: End
11: **Output:** Structure image $T$.

---

### 2.2. ITVM

Using the isotropic expression of TV term, ITVM pays more attention to the global unified detail smoothing of the image rather than structure extraction, which is widely used in image deblurring and restoration [37]. The split Bregman algorithm is an extremely efficient algorithm to solve convex optimization problems, especially ITVM, and is briefly given as follows.

ITVM wishes to solve

$$argmin_F \sum_p \left\{ \frac{\mu}{2} (F_p - R_p)^2 + \sqrt{(\nabla_x F)_p^2 + (\nabla_y F)_p^2} \right\}, \tag{9}$$

where $\mu$ is the fidelity parameter. Setting $d_x \approx \nabla_x F$ and $d_y \approx \nabla_y F$, the equation constrained optimization problem can be transformed into an unconstrained optimization problem as follows:

$$argmin_{F,dx,dy} \|d_x, d_y\|_2 + \frac{\mu}{2} \|F - R\|_2^2 + \frac{\lambda_i}{2} \|d_x - \nabla_x F - b_x\|_2^2 + \frac{\lambda_i}{2} \|(d_y - \nabla_y F - b_y)\|_2^2. \tag{10}$$

$\|d_x, d_y\|_2 = \sum_p \sqrt{d_{x,p}^2 + d_{y,p}^2}$, $\|\cdot\|_2$ denotes L-2 norm, $\lambda_i$ denotes the regularization parameter in ITVM, and $b_x$ and $b_y$ are proper constant terms. For fast iteration, we chose the Gauss–Seidel method, where the following is obtained

$$F_{ij}^{k+1} = G_{i,j}^k = \frac{\lambda}{\mu + 4\lambda} (F_{i+1,j}^k + F_{i,j+1}^k + F_{i-1,j}^{k+1} + F_{i,j-1}^{k+1}$$
$$+ d_{x,i-1,j}^k - d_{x,i,j}^k + d_{y,i,j-1}^k - d_{y,i,j}^k - b_{x,i-1,j}^k + b_{x,i,j}^k - b_{y,i,j-1}^k + b_{y,i,j}^k) + \frac{\mu}{\mu + 4\lambda} R_{i,j}. \tag{11}$$

where $i$, $j$ represent the coordinates of the pixel $p$, and $k$ denotes the number of iterations. The ITVM based on the split Bregman algorithm is given in Algorithm 2, and the comprehensive mathematical process is detailed in [43]. For simplicity, Algorithm 2 shows the process of single band image processing.

---

**Algorithm 2:** The ITVM Solution Based on Split Bregman Algorithm for Smoothing

---

1: **Input:** input image $R$; fidelity parameter $\mu$; regularization parameter $\lambda_i$;

2: **Initialize:** $F^0 = R$, and $d_x^0 = d_y^0 = b_x^0 = b_y^0 = 0$; stopping tolerance $tol = 0.1$;

3: While $\|F^k - F^{k-1}\|_2 > tol$

4: $\quad F^{k+1} = G_{i,j}^k$

5: $\quad s^k = \sqrt{\left|\nabla_x F^k + b_x^k\right|^2 + + \left|\nabla_y F^k + b_y^k\right|^2}$

6: $\quad d_x^{k+1} = \max\left(s^k - \frac{1}{\lambda}, 0\right)\frac{\nabla_x F^k + b_x^k}{s^k}$

7: $\quad d_y^{k+1} = \max\left(s^k - \frac{1}{\lambda}, 0\right)\frac{\nabla_y F^k + b_y^k}{s^k}$

8: $\quad b_x^{k+1} = b_x^k + \left(\nabla_x F^{k+1} - d_x^{k+1}\right)$

9: $\quad b_y^{k+1} = b_y^k + \left(\nabla_y F^{k+1} - d_y^{k+1}\right)$

10: End

11: **Output:** Smoothed image $F$.

---

### 2.3. Overall Design

The overall flowchart of our proposed method is given in Figure 1, and its overall calculation process is described in Algorithm 3.

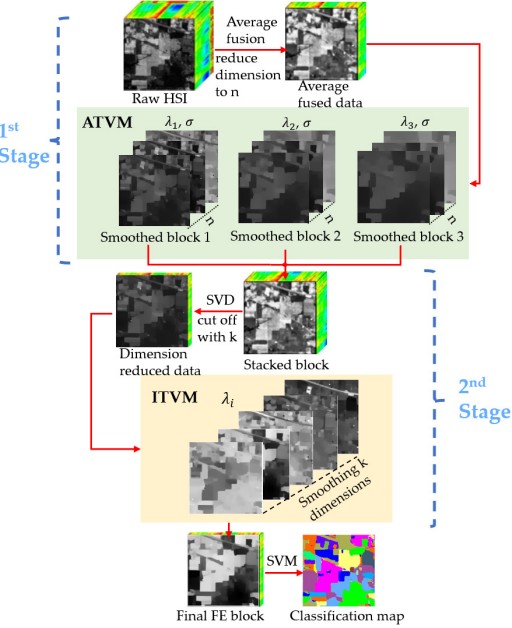

**Figure 1.** Flowchart of the proposed two-staged FE method.

The original HSI has a large amount band dimension. In order to reduce the calculation time of the subsequent steps and the influence of image noise, we first use the average fusion method for preliminary dimension reduction in the first stage. The average fusion is an effective band selecting and merging method, and more importantly, its calculation is very efficient [23]. The simple rule of average fusion method is as follows. We denote the hyperspectral data of $M$ bands as $Hi(0 < i < M)$

and split it into $N$ groups. Then the information of the $j$th band after dimension reduction can be obtained by the following fusion method ($0 < j < N$):

$$R_j = \begin{cases} mean\left(H_{(j-1)B+1}, \cdots, H_{jB}\right) & if \ jB \leq M; \\ mean(H_{N-B+1}, \cdots, H_M) & if \ jB > M; \end{cases} \quad (12)$$

Among them, $B = M/N$, and *mean* represents the calculation of each group's band dimension average value, and $B$ represents the smallest integer not greater than $M/N$.

Then at the first stage, we processed the fused HSIs based on ATVM. Inspired by [45], we used different $\lambda$ values to perform three sets of different smoothness processing, each of which uses a fixed $\sigma$ to specific the initial maximum scale size. Through this processing based on ATVM, the multi-scale structure information of the image was fully extracted. Next, we stacked these three featured blocks, used SVD to perform secondary dimension reduction at the beginning of the second stage, and then performed ITVM-based processing on each dimension using a fixed $\lambda_i$, where ITVM considered global smoothing to fully strengthen the information of all classes. The output result was classified by SVM.

---

**Algorithm 3:** The Proposed Two-Staged FE Method

---

1: **Input:** input raw image $R^M$; regularization parameter $\lambda_1, \lambda_2, \lambda_3, \lambda_i$; fidelity parameter $\mu$, scale size parameter $\sigma$; average fused number $n$, SVD number $k$;
First Stage:
2: $F^n = average\_fusion\left(R^M, n\right)$ ;
3: $F^{3n} = \left[F_1^n, F_2^n, F_3^n\right] = ATVM(F^n, \lambda_1, \lambda_2, \lambda_3, \sigma)$;
Second Stage:
4: $F^k = SVD\left(F^{3n}, k\right)$;
5: for $i = 1 : k$
6:     $T^i = ITVM\left(F_i^k, \mu, \lambda_i\right)$; $F_i^k$ denotes the $i - th$ dimension of $F^k$.
7: end
8: $T = [T^1, T^2, \cdots, T^k]$;
9: **Output:** Featured block $T$.

---

## 3. Experiments

### 3.1. Datasets

Our experiments used three real hyperspectral datasets: Indian Pines, Salinas, and Houston University 2018. Their detailed parameters are given in Table 1. In addition, the three datasets also have different characteristics. The spatial resolution and size of Indian Pines are small, but the range of its scenes is wide; Salinas covers farmland with larger image size and very uniform distribution of land objects; Houston University 2018 is a typical large size image, which is released by the IEEE GRSS 2018 data fusion contest [46,47] with high spatial resolution, various classes and scattered distribution. The full band datasets were tested, these three different datasets could fully test the effectiveness of FE methods.

**Table 1.** Details of the three experimental datasets.

| Datasets | Indian Pines | Salinas | Houston University 2018 |
|---|---|---|---|
| Source | AVIRIS sensor | AVIRIS sensor | CASI 1500 |
| Spectral Range | 0.4–2.5 μm | 0.4–2.5 μm | 0.38–1.05 μm |
| Spatial Resolution | 20 m | 3.7 m | 1 m |
| Class | 16 | 16 | 20 |
| Band | 220 | 224 | 48 |
| Spatial Size | $145 \times 145$ | $512 \times 217$ | $601 \times 2384$ |

### 3.2. Experimental Setup

#### 3.2.1. Comparison Methods

In this article, we chose some state-of-the-art methods for comparison, including:

- SVM [48]. Directly send the original datasets into SVM for classification as a basic comparison.

- Local Covariance Matrix Representation (LCMR) [49]. This is a FE method using local covariance matrices to characterize the spatial–spectral information.
- Random patches network (RPNet) [50]. This is an efficient deep learning-based method that directly regards the random patches taken from the image as the convolution kernels, which combines both shallow and deep convolutional features.
- Multi-Scale Total Variation (MSTV) [51]. This is a noise-robust method which extracts multiscale information.
- Generalized tensor regression (GTR) [52]. This is a strengthened tensorial version of the ridge regression for multivariate labels which exploits the discrimination information of different modes.
- Double-Branch Dual-Attention mechanism network (DBDA) [35]. This is a deep learning network that contains two branches, applied channel attention block and spatial attention block, to capture considerable information on spectral and spatial features.
- Fusion of Dual Spatial Information (FDSI) [29]. This is a framework using the fusion of dual spatial information, which includes pre-processing FE and post-processing spatial optimization.
- $l_0$-$l_1$ HTV [31]. This is a hybrid model that takes full consideration of the local spatial–spectral structure and yields sharper edge preservation.
- SpectralFormer [53]. This is a backbone network based on the transformer, which learns spectrally local sequence information from neighboring bands, yielding group-wise spectral embeddings.

Among them, SVM has been implemented by LIBSVM library [44] as well as a Gaussian kernel with five-fold cross-validation, where the penalty parameter c ranged from $10^{-2}$ to $10^4$ and kernel function parameter $\gamma$ ranged from $10^{-3}$ to $10^4$. In LCMR, the number of MNF principal components L was 20, the number of local neighboring pixel K was 220, and the window size T was 25. In RPNet, the number of patches k was 20 and the size of patches w was 21. In MSTV, the number of principal components used, N, was 30, the band number of the dimension-reduced was 20. In GTR, rank-1 decomposition was used. In DBDA, the batch size was set as 16, and the optimizer was set to Adam with the 0.0005 learning rate. In FDSI, the smoothing parameter $\lambda$ and the fusion weight $\mu$ were set to 1.2 and 0.5, respectively. In $l_0$-$l_1$ HTV, the sampling ratio $\mu$ was 0.1, and the parameter $\lambda_{tv}$ was 0.08. In SpectralFormer, the mini-batch size of Adam optimizer was 64, the CAF module was selected, and the learning rate was initialized with $5 \times 10^{-4}$ and decayed by multiplying a factor of 0.9 after each one-tenth of total epochs. The various main parameters mentioned above were the default values of the related algorithms that tend to achieve the highest performance. For more detailed settings and explanations, please refer to the respective references.

### 3.2.2. Experimental Parameters

In our method, according to the description in Section 2, we mainly needed to set the following parameters. In the first stage, the average fusion method was used to reduce the raw data dimension to n; n tends to affect the calculation time of subsequent processing because the larger n is, the more dimensions are retained. The regularization parameter, i.e., $\lambda_1$, controls the degree of smoothness for the image. By analyzing the smoothness of the single band image after filtering, the value ranges of three groups of different smoothness can be deduced. The scale parameter $\sigma$ specifies the initial maximum scale size and was set as the loop condition. In the second stage, the principal component k after SVD takes a value between 20 and 30, empirically [28]. The fidelity parameter $\mu$ for smoothing can tolerate a large value, and the regularization parameter in ITVM met $\lambda_i = 2\mu$, which usually results in good convergence for image processing, according to [43]. The others, such as $\varepsilon, \varepsilon_2$, and *tol*, were all set to the default initialization values as stated in Algorithms 1 and 2. Of course, the final determined value of the parameters depended on the performance; n was set to 15, $\lambda_1, \lambda_2, \lambda_3$ were set to 0.004, 0.01, 0.02, respectively, $\mu$ was set to 100, and k was set to 20. These values yielded the best performance considering the accuracy and computing time. The analysis and discussion of these main parameters will be given in detail in Section 4.

### 3.2.3. Metrics and Device

Three widely used quantitative metrics were used to evaluate the performances of all the experimental methods—average accuracy (AA) is the average of the accuracies for each class. Overall accuracy (OA) is the accuracy of each class weighted by the proportion of test samples for that class in the total training set. Kappa coefficient (Kappa) [54] calculates the percentage of the identified pixels corrected by the number of agreements, and is a measure of agreement normalized for chance agreement.

Experiments of all methods were performed on a personal computer equipped with Intel i7-9750 and NVIDIA GeForce GTX 1650. The main processor base frequency of 2.60 GHz and main memory

of 64GB ensured that all experiments were carried out easily. The software used was based on Matlab R2018b and Jupyter Notebook with PyTorch.

### 3.3. Experimental Results and Discussion

### 3.3.1. Indian Pines

To demonstrate the performance of our proposed method, a few samples were selected as the training set. For Indian Pines, we randomly selected only ten pixels from each class as the training samples and the remaining labelled pixels as the test samples. The experiments of each method were repeated ten times, and the corresponding classification results were averaged as the final value. The results are listed in Table 2, including the training set, test set, AA, OA, Kappa, and each class's accuracy. In addition, Figure 2 shows the false-color composite image, the ground truth image and the classification maps of each method in a single experiment.

**Table 2.** Experimental results (percentage) on Indian Pines dataset obtained by different methods. The best results are highlighted in bold.

| Class | Training Set | Test Set | SVM | LCMR | RPNet | MSTV | GTR | DBDA | FDSI | $l_0$-$l_1$HTV | SpectralFormer | OURS |
|---|---|---|---|---|---|---|---|---|---|---|---|---|
| 1 | 10 | 36 | 14.65 | 99.17 | 90.44 | 94.48 | **100.00** | **100.00** | 97.37 | 90.83 | 70.42 | 96.94 |
| 2 | 10 | 1418 | 42.66 | 74.92 | 68.78 | 77.80 | 66.69 | **86.36** | 81.51 | 71.27 | 48.64 | 81.73 |
| 3 | 10 | 820 | 37.52 | 73.01 | 56.83 | 79.84 | 74.32 | 80.07 | **95.08** | 76.66 | 70.50 | 79.48 |
| 4 | 10 | 227 | 15.83 | 96.65 | 41.57 | 56.73 | 93.74 | **100.00** | 88.55 | 82.38 | 82.13 | 98.59 |
| 5 | 10 | 473 | 50.74 | 88.69 | 80.53 | 92.00 | 83.51 | **99.52** | 95.46 | 81.99 | 68.87 | 88.27 |
| 6 | 10 | 720 | 78.07 | 88.24 | 94.30 | 98.69 | 94.58 | 87.80 | **99.57** | 81.79 | 89.71 | 97.15 |
| 7 | 10 | 19 | 18.34 | **100.00** | 51.59 | 71.04 | **100.00** | 65.71 | 61.14 | 95.56 | **100.00** | **100.00** |
| 8 | 10 | 468 | 90.07 | 99.59 | 91.78 | 99.98 | 96.20 | **100.00** | **100.00** | 98.21 | 98.21 | **100.00** |
| 9 | 10 | 10 | 6.86 | **100.00** | 41.04 | 65.80 | **100.00** | 93.33 | 79.05 | **100.00** | **100.00** | **100.00** |
| 10 | 10 | 962 | 36.07 | 74.31 | 68.81 | 74.93 | 69.58 | 85.82 | 78.28 | 78.87 | 77.92 | 86.14 |
| 11 | 10 | 2445 | 59.45 | 69.21 | 80.49 | 92.13 | 68.62 | 86.84 | **95.81** | 74.15 | 68.54 | 89.87 |
| 12 | 10 | 583 | 20.66 | 81.29 | 47.87 | 73.40 | 80.55 | **96.68** | 63.01 | 66.88 | 76.38 | 92.62 |
| 13 | 10 | 195 | 72.95 | 99.29 | 94.52 | 99.29 | 99.28 | **100.00** | 99.38 | 99.33 | 97.71 | 99.59 |
| 14 | 10 | 1255 | 80.94 | 96.37 | 95.80 | 99.18 | 88.37 | 91.29 | 97.28 | 96.36 | 90.69 | **99.86** |
| 15 | 10 | 376 | 35.22 | 95.29 | 59.97 | 98.53 | 94.95 | 95.73 | 94.66 | 97.13 | 46.63 | **98.88** |
| 16 | 10 | 83 | 87.57 | 97.83 | 98.10 | 95.38 | 99.28 | 93.62 | 78.26 | 92.65 | **100.00** | 93.13 |
| | AA | | 43.73 | 89.62 | 72.65 | 85.58 | 88.10 | 91.42 | 87.74 | 86.50 | 80.40 | **93.89** |
| | OA | | 46.35 | 81.17 | 70.99 | 85.75 | 80.36 | 89.08 | 88.34 | 80.75 | 73.18 | **90.64** |
| | KAPPA | | 40.10 | 78.72 | 67.40 | 83.85 | 75.69 | 87.58 | 87.24 | 78.24 | 69.67 | **89.35** |

1: Alfalfa. 2: Corn-notill. 3: Corn-mintill. 4: Corn. 5: Grass-pasture. 6: Grass-trees. 7: Grass-pasture-mowed. 8: Hay-windrowed. 9: Oats. 10: Soybean-notill. 11: Soybean-mintill. 12: Soybean-clean. 13: Wheat. 14: Woods. 15: Buildings-Grass-Trees-Drives. 16: Stone-Steel-Towers.

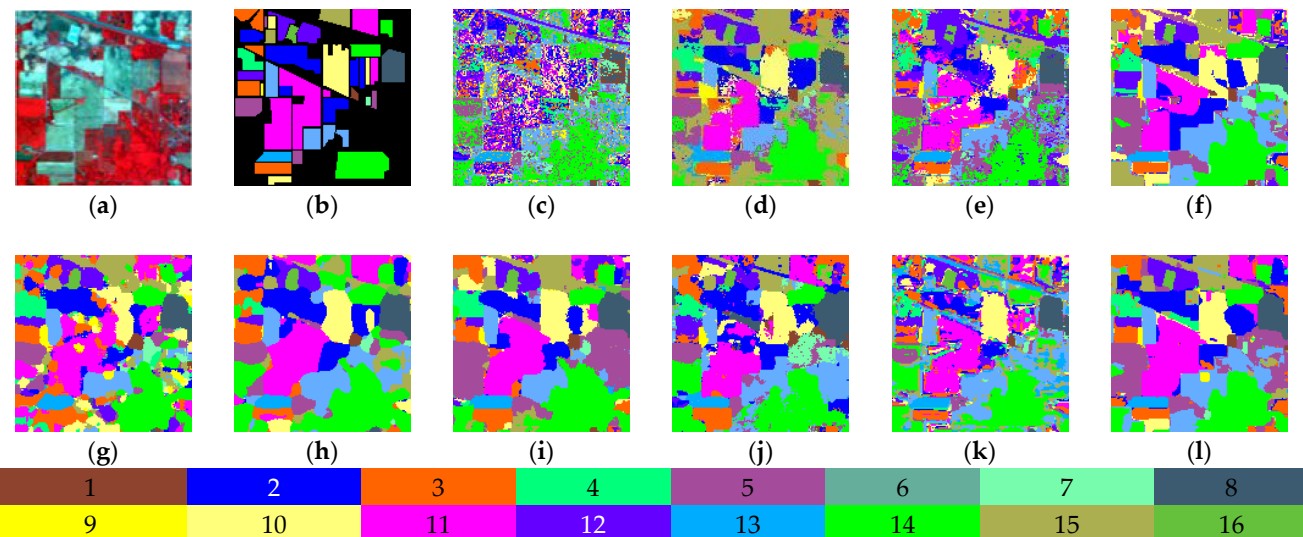

**Figure 2.** Indian Pines dataset. (**a**) False color composite image; (**b**) Ground truth image; (**c**) SVM; (**d**) LCMR; (**e**) RPNet; (**f**) MSTV; (**g**) GTR; (**h**) DBDA; (**i**) FDSI; (**j**) $l_0$-$l_1$HTV; (**k**) SpectralFormer; (**l**) Our method. Color illustrations of all classes are shown below.

As can be observed, the SVM method has the worst accuracy and kappa, and its classification map is quite noisy in the whole scene, which fully shows the importance of FE for HSI. By extracting the spatial–spectral feature information, the LCMR method significantly improved the results when compared to SVM. However, LCMR uses fixed-window clustering to obtain the covariance matrix representation information, so obvious point-like noise appears in various junction areas. Similarly, in the result classification map of GTR, there is obvious regional noise among various classes, and the boundary region was misclassified seriously. RPNet directly takes the random patch obtained from the image as the convolution kernel without any training, but when the samples are few in number, the random patch pattern seems to introduce more redundant information, and the results show obvious mixed noise in the classification map. Similarly, transformers need enough large-scale training to obtain excellent performance [55]. Although SpectralFormer is an advanced version of transformers for HSI, it still obtained a poor classification effect when the amount of training was limited, and there are many misclassifications in the classification map. In the classification map of $l_0$-$l_1$HTV, all kinds of edges are well divided, but there are obvious "corrosion"-like points inside the individual classes, which may be due to the strong constraint of the $l_0$-$l_1$ hybrid term, resulting in some internal pixel information being identified as noise, thus introducing misclassification.

MSTV and FDSI fully extract the spatial features of the image and focus on multi-scale and dual fusion and optimization on the structure information, respectively, which significantly reduced the misclassification between the class-map regions. However, the accuracy was still very low in some classes, especially for classes with few pixels, such as Oats and Grass-pasture-mowed, because these two methods aim to enhance structural features but weaken the feature representation of small sample classes, which often have much fewer edges. DBDA contains two branches to capture plenty of spectral and spatial features and yielded a great performance when the training sample was few in number, but it still had poor classification results in fewer-pixel-class Grass-pasture-mowed. In addition, it led to an obviously oversmoothed classification map. On the contrary, the proposed method performed better in the classification maps and had the highest results of OA, AA, and Kappa. A number of 14 classes in 16 had more than 86% accuracy, and 11 classes had over 90%, of which both are the highest among all methods. Especially for small sample classes, our method had 100% accuracy of oats and Grass-pasture-mowed. Due to the fact that the two-staged FE not only fully excavates the spatial structure information but also smoothly strengthen the detailed features, especially for the small sample classes, the two stages complement each other to complete the comprehensive and discriminative FE of HSI.

In addition, the time cost of each method is listed in Table 3. It is easy to see that the proposed method has the least time overhead.

**Table 3.** The computing time (seconds) of different methods for Indian Pines.

| Methods | SVM | LCMR | RPNet | MSTV | GTR | DBDA | FDSI | $l_0$-$l_1$HTV | SpectralFormer | OURS |
|---------|-----|------|-------|------|-----|------|------|------|------|------|
| Time | 4.250 | 10.021 | 2.356 | 3.452 | 4.067 | 105.61 | 7.680 | 159.59 | 331.24 | 1.354 |

### 3.3.2. Salinas

For Salinas, we randomly selected only five pixels from each class as the training samples, and the remaining samples were then used for the test. All experiments were repeated ten times, and each class's average accuracy, AA, OA, and Kappa are reported in Table 4. The classification maps of different methods are shown in Figure 3. As can be seen, the scenes in the Salinas are evenly distributed, with good separability, and the metrics of each method are higher than those in Indian Pines. Despite this, the proposed method also had the highest AA, OA, and Kappa. A number of 15 of the 16 categories had an accuracy higher than 85%, which was the best of all methods. It fully shows that our design enhances the features of global land cover, including not only edge information, but also other detailed sample features. At the same time, the noise of the classification result map is the smallest, and the classification is smooth in various junction areas, indicating that the texture information is better preserved while the edge is fully extracted.

What's more, the computing time of the eight considered methods for the Salinas image is reported in Table 5. As can be observed, although SVM has the smallest time overhead in this dataset, considering that SVM is not a FE method but a basic comparison, only the others were examined, the proposed method is the fastest one.

**Table 4.** Experimental results (%) on Salinas dataset obtained by different methods. The best results are highlighted in bold.

| Class | Training Set | Test Set | SVM | LCMR | RPNet | MSTV | GTR | DBDA | FDSI | $l_0$-$l_1$HTV | SpectralFormer | OURS |
|---|---|---|---|---|---|---|---|---|---|---|---|---|
| 1 | 5 | 2003 | 80.85 | 89.51 | 86.36 | **100.00** | 98.58 | 100.00 | 99.83 | 99.54 | 98.09 | 96.24 |
| 2 | 5 | 3721 | 95.16 | 97.41 | 97.99 | 99.67 | 92.18 | 97.25 | **100.00** | 98.97 | 94.47 | 99.81 |
| 3 | 5 | 1971 | 74.19 | 88.18 | 96.08 | 94.58 | 99.90 | 98.99 | 99.73 | 94.13 | 85.02 | **100.00** |
| 4 | 5 | 1389 | 97.16 | 95.36 | 97.83 | 93.74 | 96.68 | 93.91 | 90.04 | 99.57 | 97.82 | 98.54 |
| 5 | 5 | 2673 | 92.94 | 95.42 | 96.01 | 93.55 | 95.31 | 93.63 | 94.69 | 91.91 | **97.37** | 96.65 |
| 6 | 5 | 3954 | **100.00** | 98.48 | 99.88 | 99.16 | 99.82 | 99.85 | 100.00 | 95.19 | 99.04 | 99.67 |
| 7 | 5 | 3574 | 94.12 | 93.77 | 93.78 | 99.61 | **99.62** | 97.89 | 97.91 | 98.70 | 95.53 | 97.64 |
| 8 | 5 | 11,266 | 64.31 | 74.82 | 67.17 | 84.01 | 68.48 | 77.19 | **94.38** | 71.04 | 46.08 | 82.23 |
| 9 | 5 | 6198 | 95.45 | 99.08 | 99.08 | 97.29 | 99.99 | 97.46 | 99.37 | 99.74 | 97.04 | **100.00** |
| 10 | 5 | 3273 | 66.16 | 93.32 | 72.47 | 93.26 | 90.89 | 95.61 | **96.24** | 88.65 | 73.27 | 92.10 |
| 11 | 5 | 1063 | 61.36 | 94.40 | 85.38 | 96.87 | 98.52 | 88.00 | 73.41 | 94.37 | 88.45 | **99.61** |
| 12 | 5 | 1922 | 77.38 | 98.51 | 91.34 | 93.68 | 94.34 | 100.00 | 94.66 | 100.00 | 64.71 | 99.54 |
| 13 | 5 | 911 | 80.09 | 95.13 | 86.22 | 97.25 | 97.65 | 96.38 | **99.96** | 97.89 | 97.77 | 93.15 |
| 14 | 5 | 1065 | 67.62 | 96.50 | 88.22 | 90.18 | 90.93 | 97.44 | 94.0 | 94.72 | 89.90 | **99.19** |
| 15 | 5 | 7263 | 43.18 | **87.18** | 47.74 | 70.95 | 69.36 | 81.69 | 67.87 | 86.76 | 72.39 | 85.83 |
| 16 | 5 | 1802 | 94.68 | 91.18 | 76.56 | 98.67 | 99.13 | 100.00 | 100.00 | 85.37 | 86.23 | 99.91 |
| | AA | | 80.85 | 93.58 | 86.38 | 93.90 | 93.21 | 94.70 | 93.88 | 93.53 | 86.45 | **96.26** |
| | OA | | 74.58 | 88.65 | 80.50 | 89.50 | 87.33 | 90.88 | 90.39 | 89.75 | 79.33 | **93.28** |
| | KAPPA | | 71.96 | 86.93 | 78.39 | 88.33 | 85.94 | 89.82 | 89.35 | 88.65 | 77.16 | **91.84** |

1: Brocoli_green_weeds_1. 2: Brocoli_green_weeds_2. 3: Fallow. 4: Fallow_rough_plow. 5: Fallow_smooth. 6: Stubble. 7: Celery. 8: Grapes_untrained. 9: Soil_vinyard_develop. 10: Corn_senesced_green_weeds. 11: Lettuce_romaine_4wk. 12: Lettuce_romaine_5wk. 13: Lettuce_romaine_6wk. 14: Lettuce_romaine_7wk. 15: Vinyard_untrained. 16: Vinyard_vertical_trellis.

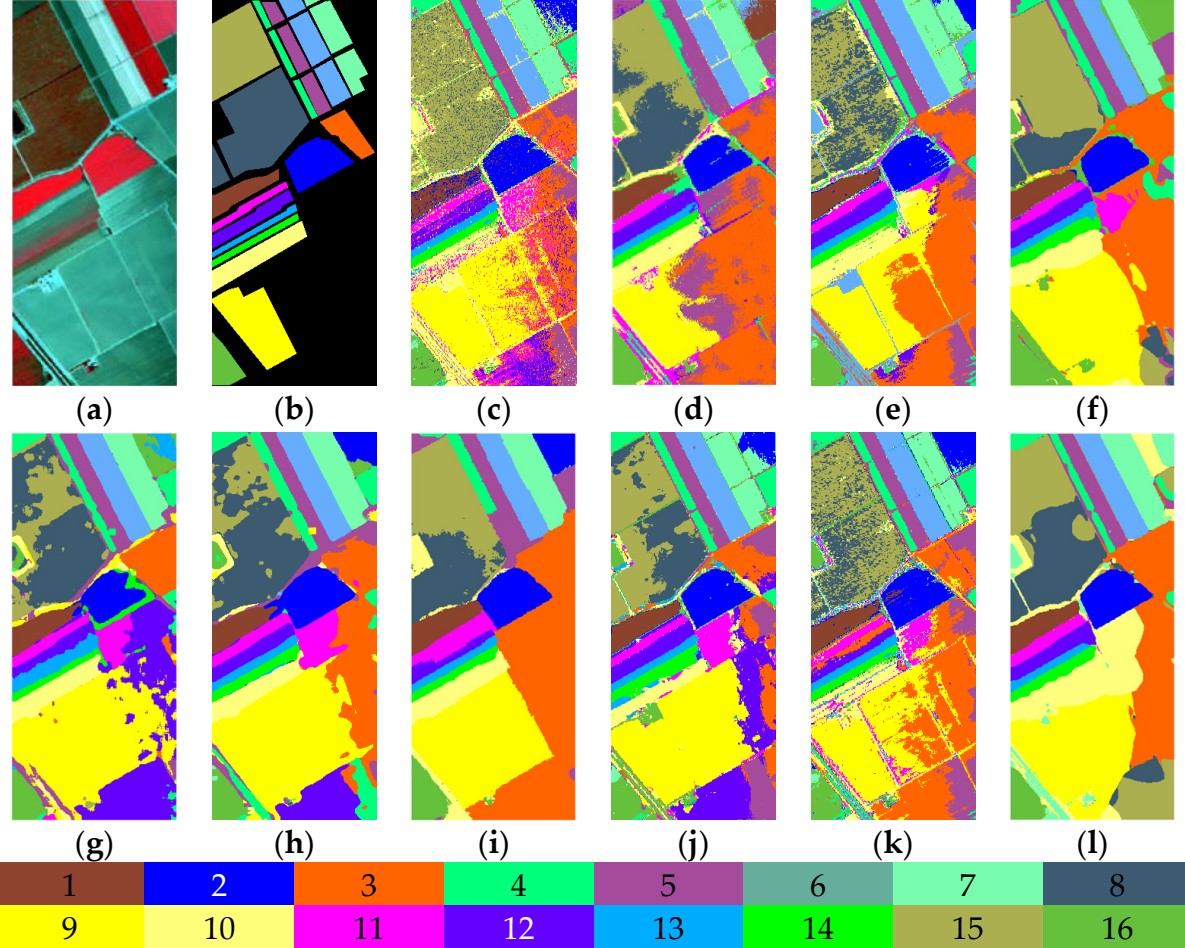

**Figure 3.** Salinas dataset. (**a**) False color composite image; (**b**) Ground truth image; (**c**) SVM; (**d**) LCMR; (**e**) RPNet; (**f**) MSTV; (**g**) GTR; (**h**) DBDA; (**i**) FDSI; (**j**) $l_0$-$l_1$HTV; (**k**) SpectralFormer; (**l**) Our method. Color illustrations of all classes are shown below.

**Table 5.** The computing time (seconds) of different methods for Salinas.

| Methods | SVM | LCMR | RPNet | MSTV | GTR | DBDA | FDSI | $l_0$-$l_1$HTV | SpectralFormer | OURS |
|---------|-----|------|-------|------|-----|------|------|----------------|----------------|------|
| Time | 4.32 | 57.77 | 10.23 | 15.94 | 15.61 | 334.94 | 33.15 | 857.75 | 1686.43 | 6.45 |

### 3.3.3. Houston University 2018

The third experiment was performed on the Houston University 2018. Due to the large size of this dataset, 100 pixels were selected for training, and the remaining were used as evaluation samples. Similarly, each group of experiments was repeated ten times and averaged. The detailed numbers of training and test samples and all the classification results are given in Table 6, and the corresponding classification maps are shown in Figure 4. It can be seen that the proposed method is also the highest in AA and Kappa. It is worth mentioning that, in the Houston dataset, the samples of community class accounted for about 44% of the total. DBDA performed well in the community class, so its OA was 1.33% higher than our method, but its AA was 4.8% lower than ours. Both LCMR and ours had 11 classes with more than 90% accuracy, but the OA of LCMR was far weaker because LCMR performed much poorer in the community class. Overall, our algorithm performed better in more classes, which proves that our algorithm improves the discrimination ability of different ground objects. However, our method has a poor classification effect in Road and Sidewalk, and the performance of other approaches in these two classes was also lower than their average level. These two classes show the characteristics of "slender and long" land cover, possibly because this feature is easily weakened in the process of FE. How to solve this problem is a topic worth discussing.

**Table 6.** Experimental results (percentage) on Houston University 2018 dataset obtained by different methods. The best results are highlighted in bold.

| Class | Training Set | Test Set | SVM | LCMR | RPNet | MSTV | GTR | DBDA | FDSI | $l_0$-$l_1$HTV | SpectralFormer | OURS |
|-------|--------------|----------|-----|------|-------|------|-----|------|------|----------------|----------------|------|
| 1 | 100 | 9699 | 80.30 | 87.28 | 68.30 | 61.53 | 91.65 | 83.86 | 52.36 | 83.89 | **97.46** | 84.02 |
| 2 | 100 | 32,402 | 85.35 | 84.12 | 85.51 | **88.23** | 54.90 | 85.26 | 83.90 | 73.30 | 84.93 | 79.16 |
| 3 | 100 | 584 | 95.60 | **100.00** | 23.74 | 95.71 | **100.00** | **100.00** | **100.00** | 99.77 | **100.00** | **100.00** |
| 4 | 100 | 13,488 | 76.84 | 96.95 | 80.43 | 79.30 | **98.11** | 92.88 | 74.79 | 90.31 | 95.25 | 95.39 |
| 5 | 100 | 4948 | 26.93 | 93.72 | 26.24 | 44.39 | 86.90 | 79.91 | 50.70 | 79.92 | 82.50 | **95.56** |
| 6 | 100 | 4416 | 42.54 | 99.68 | 61.35 | 93.82 | **100.00** | 98.16 | 60.33 | 99.44 | 95.81 | **100.00** |
| 7 | 100 | 166 | 51.37 | **100.00** | 66.94 | 95.54 | **100.00** | **100.00** | 97.24 | 98.39 | 99.40 | 99.04 |
| 8 | 100 | 39,662 | 51.62 | 84.73 | 65.14 | 75.01 | 70.51 | 79.38 | 85.76 | 88.70 | 65.15 | **92.76** |
| 9 | 100 | 223,584 | 96.24 | 71.56 | 95.72 | **97.79** | 56.52 | 92.51 | 92.04 | 70.50 | 62.30 | 89.39 |
| 10 | 100 | 45,710 | 48.33 | 50.96 | 49.63 | 66.27 | 15.29 | 62.43 | **72.83** | 42.56 | 32.20 | 58.33 |
| 11 | 100 | 33,902 | 42.62 | 48.63 | 40.70 | 48.93 | 6.40 | **63.06** | 54.39 | 29.07 | 29.52 | 49.31 |
| 12 | 100 | 1416 | 4.54 | 76.91 | 5.76 | 9.71 | 12.50 | 35.62 | 12.80 | 73.33 | 45.83 | **85.00** |
| 13 | 100 | 46,258 | 59.19 | 55.57 | 69.27 | 82.14 | 30.90 | 69.78 | **86.11** | 61.13 | 35.25 | 69.88 |
| 14 | 100 | 9749 | 42.37 | 94.26 | 72.05 | 82.81 | 97.81 | 83.91 | 77.01 | 96.38 | 84.76 | **98.99** |
| 15 | 100 | 6837 | 60.40 | **99.60** | 66.39 | 94.96 | 92.69 | 91.87 | 91.04 | 95.58 | 96.31 | 98.51 |
| 16 | 100 | 11,375 | 56.61 | **89.47** | 55.27 | 82.77 | 41.40 | 82.86 | 83.44 | 67.27 | 70.58 | 84.88 |
| 17 | 100 | 49 | 4.65 | **100.00** | 2.46 | 40.20 | **100.00** | 84.94 | 44.89 | **100.00** | **100.00** | **100.00** |
| 18 | 100 | 6478 | 23.36 | **90.57** | 33.63 | 66.66 | 85.88 | 86.01 | 67.40 | 69.79 | 75.75 | 81.62 |
| 19 | 100 | 5265 | 30.00 | **97.43** | 53.70 | 80.18 | 80.58 | 90.81 | 66.64 | 81.81 | 85.38 | 92.05 |
| 20 | 100 | 6724 | 56.75 | **99.90** | 52.86 | 75.35 | 99.52 | 94.55 | 89.55 | 99.01 | 95.17 | 99.85 |
| AA | | | 51.78 | 86.07 | 53.76 | 73.07 | 71.08 | 82.89 | 72.16 | 80.01 | 76.58 | **87.69** |
| OA | | | 62.88 | 72.15 | 67.37 | 80.90 | 52.61 | **83.49** | 80.37 | 68.37 | 59.67 | 82.16 |
| KAPPA | | | 55.86 | 66.00 | 60.80 | 75.90 | 44.88 | 77.13 | 76.19 | 61.57 | 52.19 | **77.27** |

1: Healthy grass. 2: Stressed grass. 3: Synthetic grass. 4: Evergreen trees. 5: Deciduous trees. 6: Soil. 7: Water. 8: Residential. 9: Commercial. 10: Road. 11: Sidewalk. 12: Crosswalk. 13: Major thoroughfares. 14: Highway. 15: Railway. 16: Paved parking lot. 17: Gravel parking lot. 18: Cars. 19: Trains. 20: Seats.

Similarly, the computing time is given in Table 7. When the amount of data increased to the dimension such as those seen in Houston University 2018 image, the running time of each method was significantly different. In LCMR, the calculation of the covariance matrix feature of each pixel consumes a lot of time, and manifold-based distance metric takes up plenty of storage space. In MSTV, the structural features require more loops, and the kernel method is also time-consuming. GTR also requires more calculations of ridge regression. Dual fusion framework in FDSI obviously requires more computing time because of the pre-post-processing. In $l_0$-$l_1$HTV, the optimization formula of the $l_0$-$l_1$ regularization terms need to divide into five subproblems, which, individually, are constrained minimization problems that require many iterate computations. As for the deep learning method, RPNet has a simpler architecture and was less time-consuming when compared with other networks, but it still consumes more time to calculate the convolution of random patches

in each layer, especially for large size images. In DBDA, selecting small pieces from the original cube data cost a considerable amount of time. SpectralFormer needs hundreds of epochs to reach a good performance.

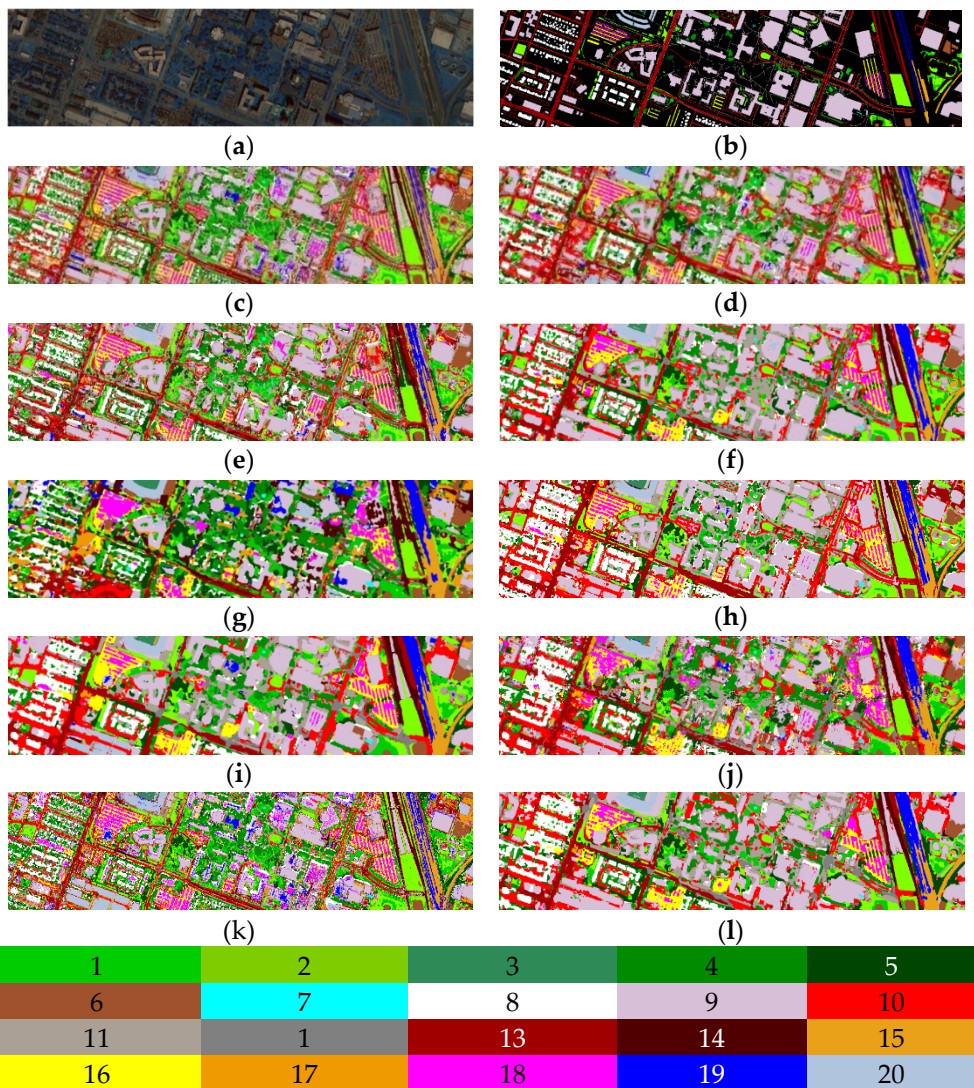

| 1 | 2 | 3 | 4 | 5 |
|---|---|---|---|---|
| 6 | 7 | 8 | 9 | 10 |
| 11 | 1 | 13 | 14 | 15 |
| 16 | 17 | 18 | 19 | 20 |

**Figure 4.** Houston University 2018. (**a**) False color composite image; (**b**) Ground truth image; (**c**) SVM; (**d**) LCMR; (**e**) RPNet; (**f**) MSTV; (**g**) GTR; (**h**) DBDA; (**i**) FDSI; (**j**) $l_0$-$l_1$HTV; (**k**) Spectral-Former; (**l**) Our method. Color illustrations of all classes are shown below.

**Table 7.** The computing time (seconds) of different methods for Houston University 2018.

| Methods | SVM | LCMR | RPNet | MSTV | GTR | DBDA | FDSI | $l_0$-$l_1$HTV | SpectralFormer | OURS |
|---------|-----|------|-------|------|-----|------|------|------------|----------------|------|
| Time | 198.08 | 522.71 | 445.60 | 226.44 | 233.04 | 22,202.63 | 722.46 | 2743.76 | 6524.85 | 75.71 |

The time cost of our method mainly comes from the processing of ATVM and ITVM. When inputting an HSI $R^{r*c*b}$, $r$ and $c$ are spatial dimensions, and $b$ is the spectral dimension. The time complexity of ATVM is $O(rc)$, and the time complexity of ITVM is $O(rc/\ln rc)$, both of which are less time-consuming. For comparison, the time complexity of $l_0$-$l_1$HTV is $O(rcb + z + rc\log rc)$, and the calculation time was far higher than ours. Intuitively, In ATVM, the scale size parameter acting as the loop condition can control the number of the iteration within a few. In ITVM, the large value of the fidelity parameter can greatly reduce the number of loops. The average fusion method only involves the average sum operation, and SVD only decomposes one matrix, therefore, both of them

consume quite little time. Overall, our method had the shortest computing time, which improves the real-time application of HSI. Moreover, our classification performance is the best.

## 4. Parameter Analysis and Discussion

### 4.1. First Stage

In the first stage, the parameter n was first investigated. In light of the characteristics of average fusion, n ranged from seven to $2/M$ (i.e., half the spectral dimension of the image), where seven could ensure that the stacked features can complete the SVD in the second stage, which was $3 \times 7 > 20$. The other parameters were fixed, with default values as stated in 3.2. The same controlled variable method was used in the following experiments. All three datasets were tested, and their results are shown in Figure 5.

As can be seen, with the increase of n, the classification results show a downward trend in the Indian Pines, but a slight increase after n is greater than 90. The decline is much smaller in Salinas and basically unchanged in Houston University 2018, especially for AA. Although there are certain fluctuations in the trends, two points can be determined: First, the highest accuracies of the three datasets were all basically achieved with a small n, and secondly, the time cost increases steadily with the increase of n, especially in Houston University 2018. Therefore, n was set to 15 for our design. Qualitatively, when n is small, more band information is fused to make the final performance better. The results of Houston University 2018 dataset have little change because its spectral dimension was low, only 48, which weakens the role of average fusion. The smaller n is, the smaller the spectral dimension that is retained, naturally reducing the subsequent processing time. Moreover, the average fusion itself has high computational efficiency. In conclusion, the average fusion method in our design is the beginning of efficient processing.

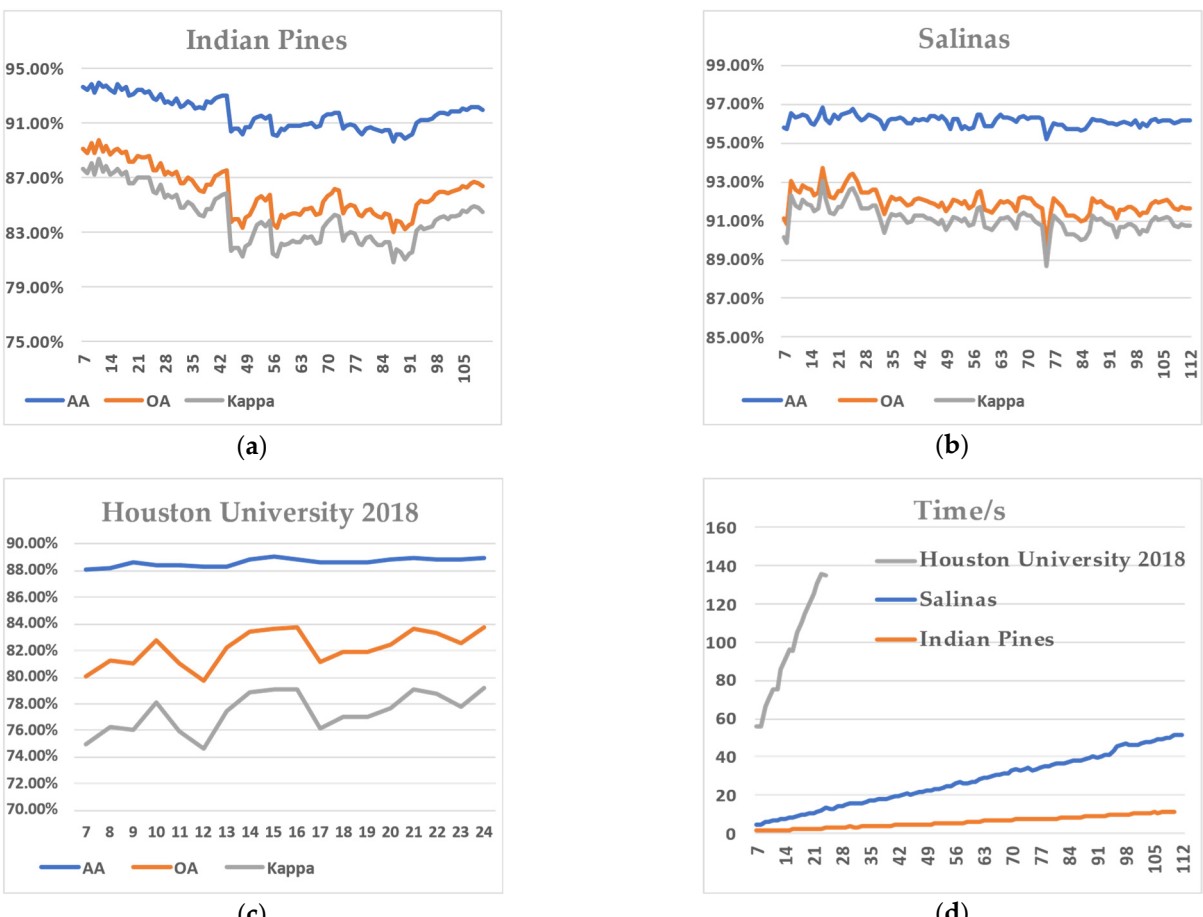

**Figure 5.** Effect of using different n in three datasets. (**a**) Indian Pines, with n growing from 7 to 110; (**b**) Salinas, with n growing from 7 to 112; (**c**) Houston University 2018, with n growing from 7 to 24. (**d**) Computing time in corresponding datasets.

In the first stage, the second type of parameters, regularization parameter $\lambda$ and scale size parameter $\sigma$, were then investigated. In the three feature blocks to be stacked, we used the same $\sigma$ value, and then we set $\sigma$ from one to six to observe the changes in the classification results shown in Figure 6.

It can be seen that according to our design, when $\sigma$ increases, there are additional loops, which expands the time overhead, and when $\sigma$ is large, the classification effect decreases. When is $\sigma = 2$, our design yielded the best classification results in the three datasets. This means that when $\sigma$ is two, ITVM in our design can efficiently extract multi-scale information, especially considering that the three datasets had a completely different scale and spatial resolution. This shows that this parameter in our design adapts to a wide range of data types and is robust. Moreover, when $\sigma$ is small, the loop is naturally reduced, and the calculation time was saved. But $\sigma$ from one to two increased the computing time, so setting $\sigma$ to two was a trade-off for accuracy.

Regularization parameters can control the degree of smoothness, which is an important part of the optimization model. Our design used three different sets of regularization parameters, i.e., $\lambda_1 = 0.004, \lambda_2 = 0.01, \lambda_3 = 0.02$, which are the only different values that were set in the first stage. We conducted the experiments with Indian Pines. Figure 7 gives the featured blocks which output by ATVM under different parameters. For comparison, the smoothed block after ITVM under $\lambda_i$ in the second stage is also shown here.

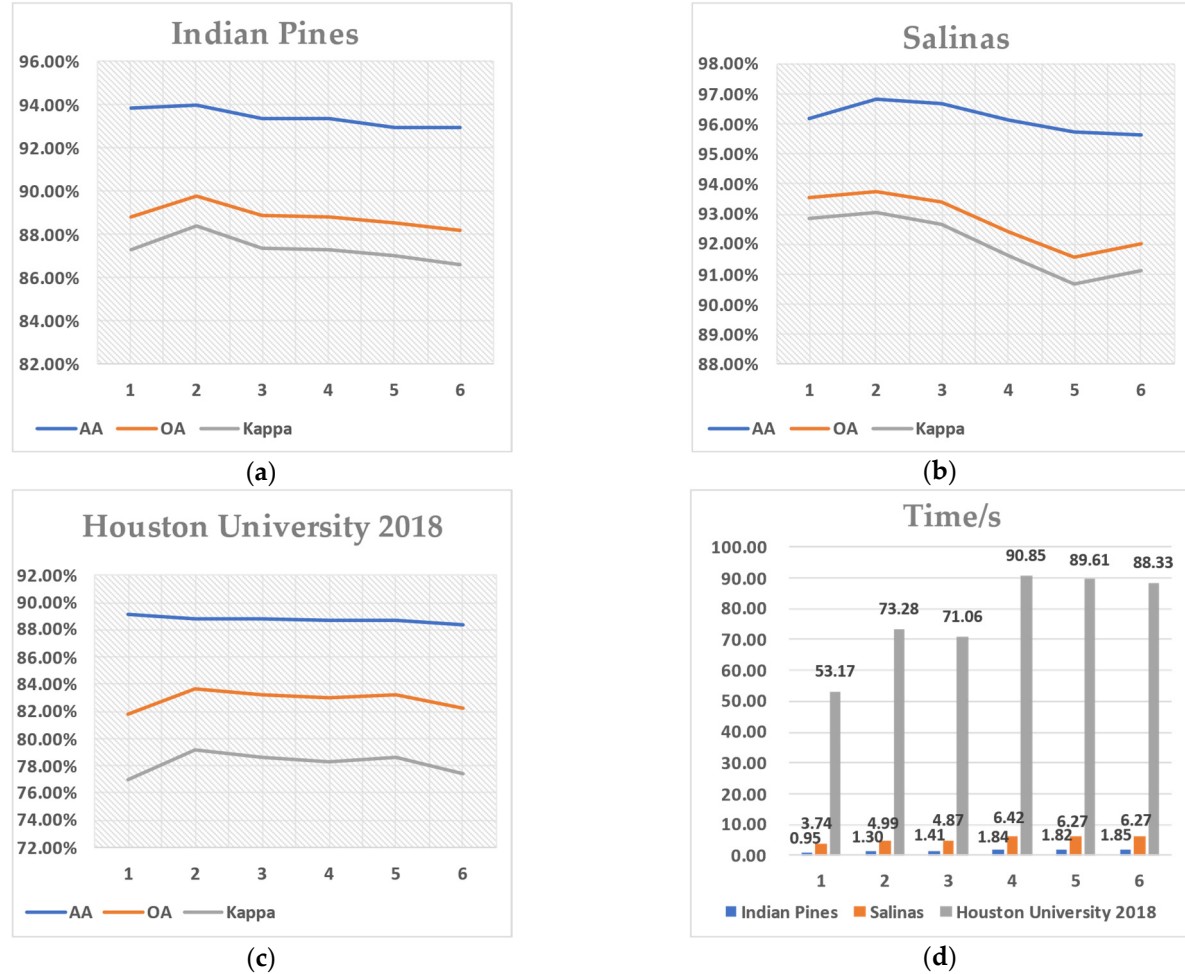

**Figure 6.** Effect of using different $\sigma$ in three datasets. (**a**) Indian Pines, (**b**) Salinas, (**c**) Houston University 2018, and (**d**) computing time in corresponding datasets.

As can be observed, when $\lambda_1 = 0.004$, the information of various land cover is clear, $\lambda_3 = 0.02$, and only obvious edges are retained. With the increase of $\lambda$, the single band images are oversmoothed and more blurred. Therefore, we chose three typical $\lambda$ values to represent three different degrees of smoothness. Besides, it can be seen from (d), in the single-band image after ITVM in the second

stage, that the structure feature was strengthened, and the detailed information is better preserved. Figure 7 shows the intuitive judgment. In the ATVM, the multi-scale structure information with different smoothness was extracted, but the information of some samples with few pixels seems to be weakened. However, through the ITVM in the second stage, the representation of each feature information was strengthened, which intuitively shows the effectiveness of our two-staged design.

We selected five representative values at the three smoothing degrees, with an interval of 0.001, and the classification results are shown in Figure 8. The values we chose yielded the best results, and they were set as the default parameters. The range of smoothing parameters is wide, so we chose representative values of three different smoothness degrees. The experimental results show that they have the best results in the corresponding interval. Additionally this setting can yield satisfactory classification accuracies for different datasets, as shown in Section 3.

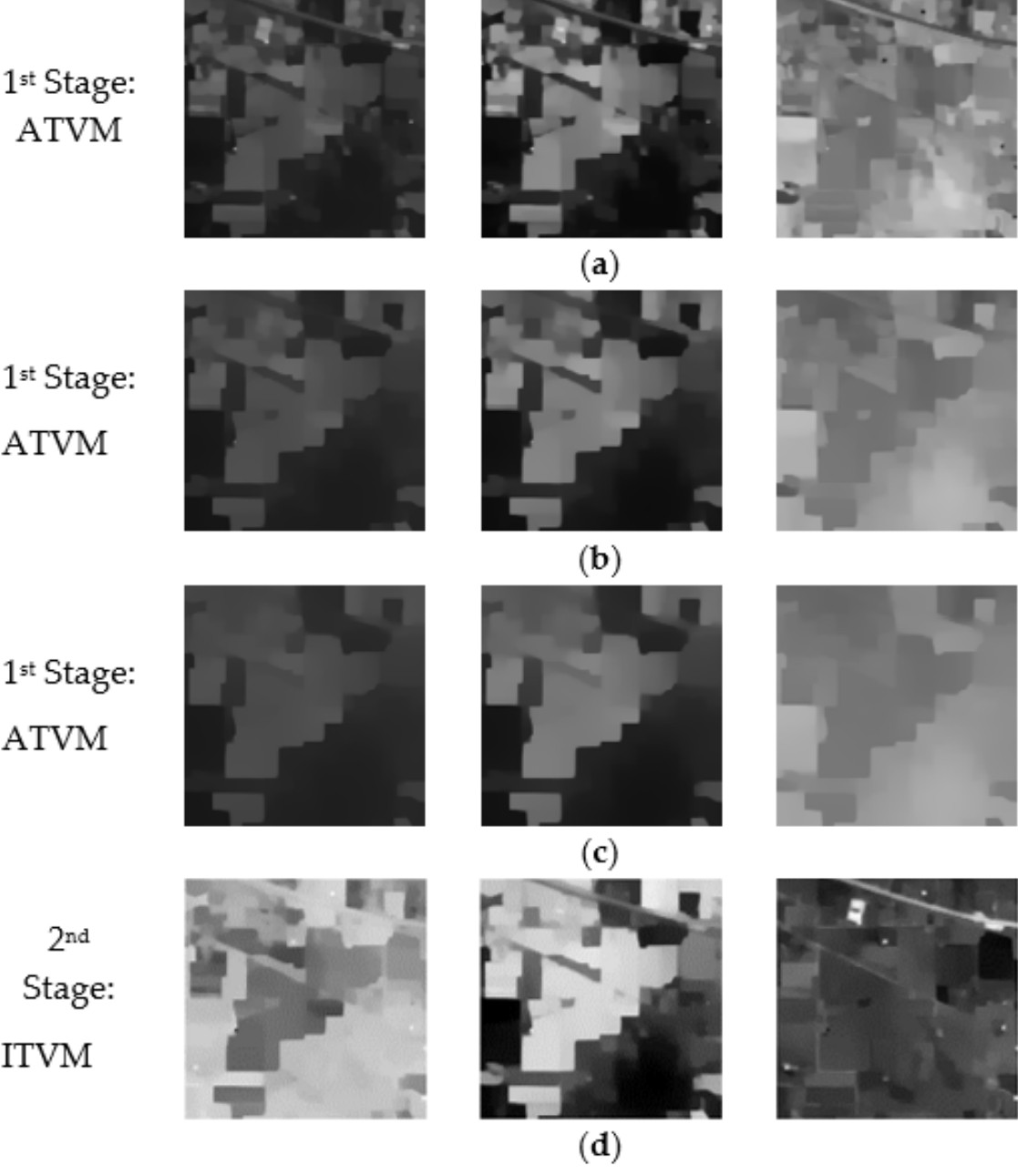

**Figure 7.** Indian Pines dataset. The results of models under different parameters, which (**a**) $\lambda_1 = 0.004$, (**b**) $\lambda_1 = 0.01$, (**c**) $\lambda_3 = 0.02$, (**d**) $\lambda_i = 100$. Each row represents the first, the third, and the fifth dimension single-band image of the corresponding results from left to right.

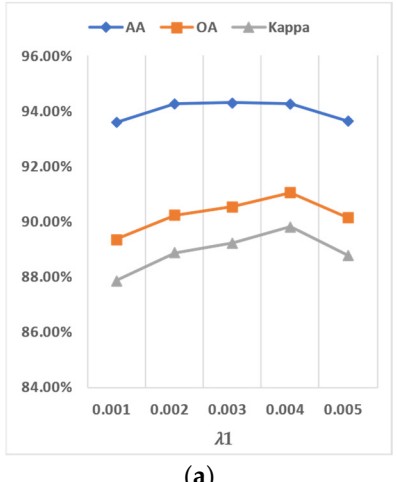 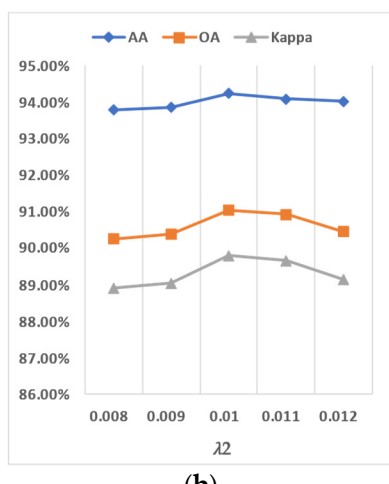 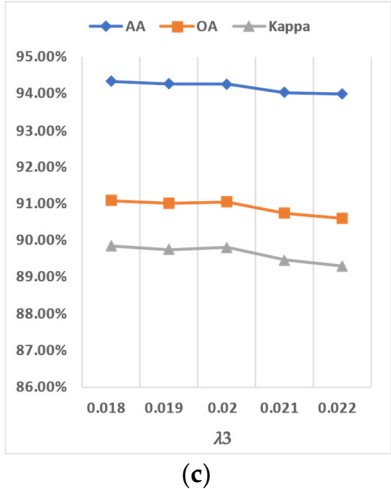

(**a**)            (**b**)            (**c**)

**Figure 8.** Analysis of the sensitivity of different $\lambda$ in the first stage based on Indian Pines datasets. We selected five representative values at the three smoothing degrees, with an interval of 0.001. (**a**) $\lambda_1$ ranges from 0.001 to 0.005; (**b**) $\lambda_2$ ranges from 0.008 to 0.012; (**c**) $\lambda_3$ ranges from 0.018 to 0.022.

### 4.2. Second Stage

There are two main parameters in the second stage: the number k of the principal component after SVD and the fidelity parameter $\mu$. Similarly, k ranged from 10 to 40 with an interval of two, with regard to of the dimension of the stacked feature blocks being 45 in the first stage. As shown in Figure 9, with the increase of k, the classification performance improved stably on the three datasets. When k was larger than 20, the metrics decreased in Indian Pines, Salinas remained stable, and there was a slight improvement in Houston University 2018. We set 20 as the default k value. Qualitatively, When the size of the image is large, more principal components act to enhance the feature information so that the performance will improve. Still, when the image size is too small, such as the Indian Pines, more principal components will bring redundant information and reduce accuracy. In addition, the time overhead is mainly concentrated in the SVD calculation, and it steadily expands with the increase of k. In summary, SVD can effectively extract the principal component information and reduce the subsequent calculation time, playing a key role in our design. Parameter settings show stability in three datasets, especially for large-scale images.

The fidelity parameter $\mu$ is very tolerant of large values according to [35]. We firstly performed experiments to verify that this conclusion is also applicable to different HSIs. In our experiments, $\mu$ ranged from 2 to 10 with step two and grew from 10 to 150 with step 10. The classification results of the three datasets are listed in Figure 10. When $\mu$ increased to 10, the results were significantly improved. When $\mu$ was greater than 20, the performance improvement in the three datasets became slow and entered the stable zone. In addition, when the value of $\mu$ was large, the performance did not fluctuate greatly and did not show a downward trend. It can be obtained that $\mu$ can tolerate large values in different characteristics of HSI. From the formula, the larger the value, the stricter the constraint on the fidelity term, the more delicate the smoothing of each feature, and there will be no oversmoothed situation. We set as the default value of the proposed method. Parameter $\mu$ is an important parameter in the second stage. Once again, the performance in three datasets fully proves that our design is robust and stable.

Finally, the results of sending the output feature blocks of the first and second stages directly to the classifier are shown in Figure 11. As a comparison, the results of feeding the raw data and the average fused data to the classifier are also listed in this Figure. This experiment demonstrated that the two stages' designs contributed significantly to classification performance in the three datasets. The results here more fully prove that our two-stage design is effective. Each stage plays an important role and greatly improves the effect.

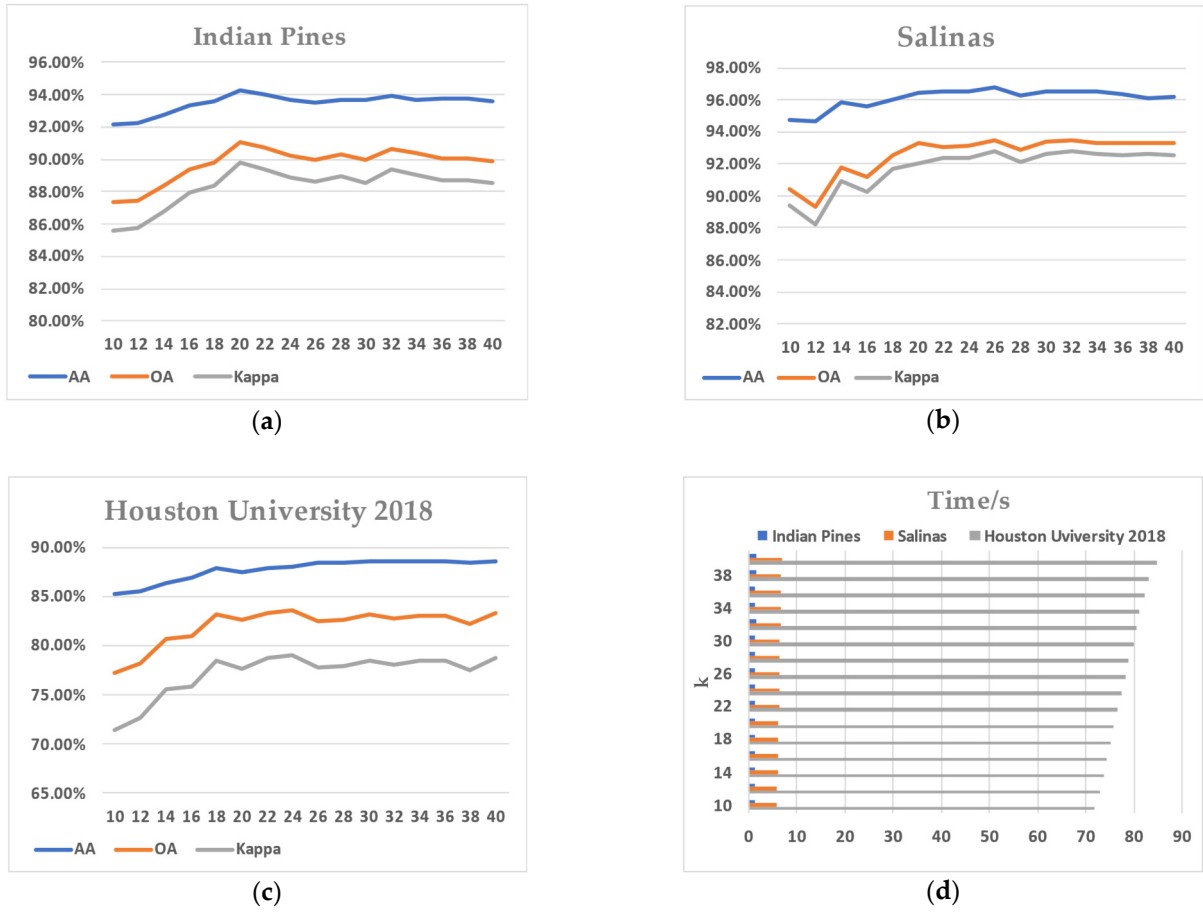

**Figure 9.** Sensitivity analysis of parameter k in the second stage. k ranges from 10 to 40 with step 2 and the classification results in (**a**) Indian Pines; (**b**) Salinas; (**c**) Houston University 2018, and (**d**) computing time in corresponding datasets.

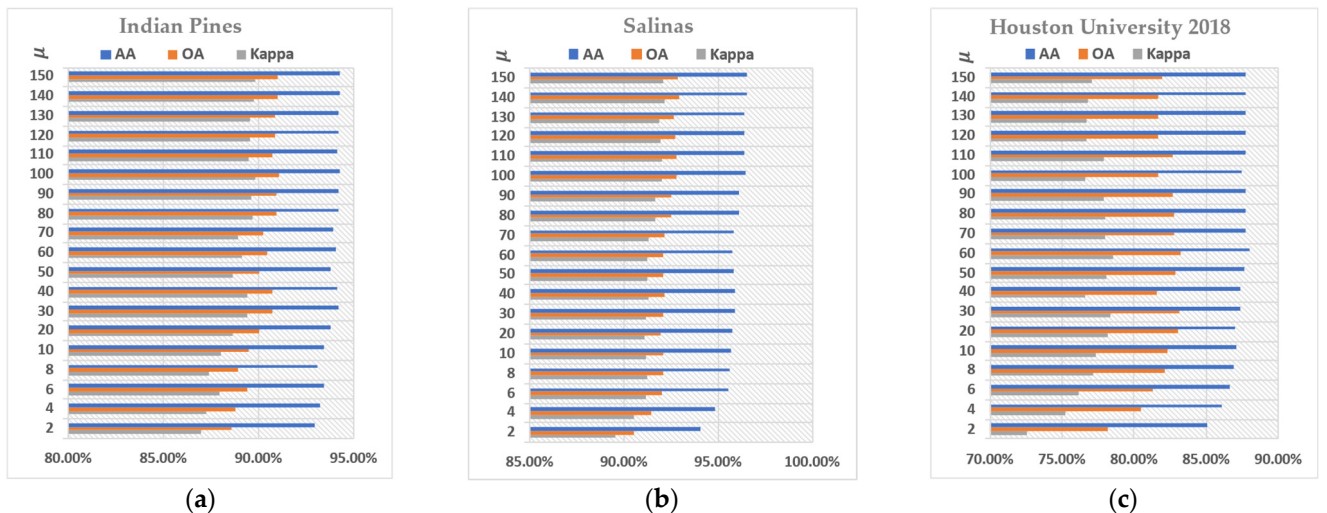

**Figure 10.** Analysis of the sensitivity of parameter $\mu$ in the second stage for three datasets. Classification results: (**a**) Indian Pines; (**b**) Salinas; (**c**) Houston University 2018.

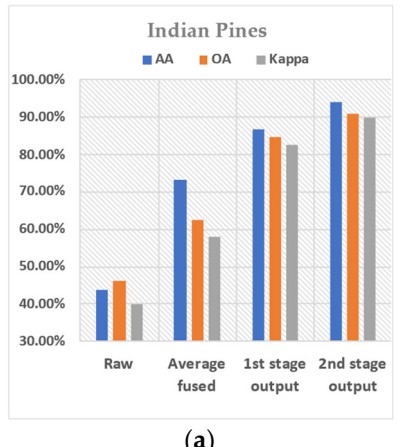 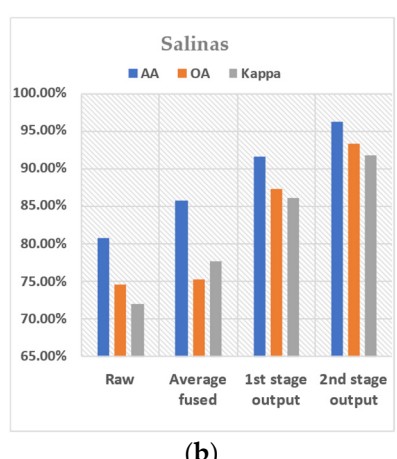 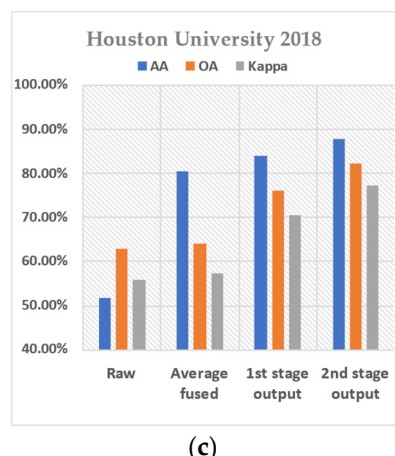

(**a**)　　　　　　　　　　　(**b**)　　　　　　　　　　　(**c**)

**Figure 11.** Classification performance of the SVM on the raw data, the average fusion data, and the output feature blocks of the first and second stages for the three datasets. (**a**) Indian Pines; (**b**) Salinas; (**c**) Houston University 2018.

## 5. Conclusions

To improve classification performance and reduce the time cost, this article innovatively proposes an efficient two-staged FE method based on TV for HSI. Based on different solutions of anisotropic and isotropic models, it successively completed the extraction of multi-scale structure information and detail smoothing enhancement of HSI, yielding distinguished classification performance. In addition, this design has no complex framework nor a redundant loop, which greatly reduces computational overhead. When compared with many state-of-the-art algorithms, our method can significantly outperform others in classification performance and computing time, especially achieving better results in most classes. More importantly, we give a sufficiently detailed parameter analysis and give the reasonable value and change explanation of each parameter from algorithm design and performance. The results show that our method has strong robustness and stability in various datasets, which further strengthens the advantages in hyperspectral practical application.

In the future, we will try to improve the classification performance of the "slender and long" land cover distribution. Although our default parameters are applicable to the three datasets, adaptive selection of parameters for different datasets is still an important improvement direction. In addition, we still have some trade-offs between time and performance, giving its parallel implementation is the primary key, especially considering the future development of HSI towards larger size and higher resolution. Finally, the practical application of HSIs is always a topic of discussion, which will face more environmental factors and noise in the future. We can use the excellent solutions and ideas for dealing with complex noise [56,57] for traditional images and transplant it to hyperspectral applications.

**Author Contributions:** C.L., developed the design and methodology, implemented the algorithm, executed the experiments, led the writing of the manuscript; X.T., assisted algorithm migration and verification; L.S. assisted several experiments; Y.T., revised the manuscript. Y.T. and Y.P., administrated the project and supervised the research. All authors have read and agreed to the published version of the manuscript.

**Funding:** This work was partially supported by the National Natural Science Foundation of China (No. 91948303-1, No. 61803375, No. 12002380, No. 62106278, No. 62101575, No. 61906210) and the National University of Defense Technology Foundation (No. ZK20-52).

**Institutional Review Board Statement:** Not applicable.

**Informed Consent Statement:** Not applicable.

**Data Availability Statement:** The datasets involved in this paper are all public.

**Acknowledgments:** The authors acknowledge State Key Laboratory of High Performance Computing, College of Computer Science and Technology, National University of Defense Technology, China. More, the authors would like to thank the IEEE GRSS Image Analysis and Data Fusion Technical

Committee and the Hyperspectral Image Analysis Lab at the University of Houston for distributing online the Houston 2018 dataset used in this paper.

**Conflicts of Interest:** The authors declare no conflict of interest.

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
