# Peer review of "A Two-Staged Feature Extraction Method Based on Total Variation for Hyperspectral Images"

_remotesensing, doi:10.3390/rs14020302_

Round 1

Reviewer 1 Report

Referee’s Report on the paper:

A Two-Staged Feature Extraction Method Based on Total Variation for Hyperspectral Images

by Chunchao Li, Xuebin Tang, Lulu Shi, Yuanxi Peng, Yuhua Tang

submitted to Remote Sensing

General comments

The paper deals with the tests on real datasets of a method, developed by the Authors, for the effective feature extraction to be applied to hyperspectral image (HSI) processing. In particular, satellite data have been taken into account.

This subject is of interest for Remote Sensing.

The paper is orderly, corrected and properly presented. Data are clearly reported and figures are appropriate.

I have only minor comments.

The Authors take for granted that they deal with satellite HIS. However, the method could be used for all kinds of hyperspectral data, even if regarding mapping at smaller scales and not obtained by passive but by active techniques for remote sensing.

Then, on one hand They could comment this possibility and enlarge the field of application, on the other hand They should better explain and introduce in the abstract and in the introduction their specific field of application. The first section where it is explained They will work with satellite data is the experimental part (section 3.1).

Detailed comments

The Authors affirm (line 32) that “…HSI itself has certain defects. Firstly, its spatial resolution is not high…”. The statement lacks of basis and absolute references. The spatial resolution depends on the specific sensor (among other things, even the examples brought by the Authors themselves have very different spatial resolutions) and the quality of the spatial resolution is always to be referred to the specific sample acquired, to its characteristic and to the information sought.

In Table 1, the number of bands/channel of the sensor should be made explicit (moreover the methods tested are applied to all the bands available or just to a restricted selection?).

The Authors should better make explicit that the classification regards land use (also here the application is taken for granted and it appears just in table 2, in the section 3.3).

The kappa coefficient discussed is the Cohen kappa?

I suggest to give short recall on the three key quantitative metrics used for the whole discussion of the experimental results.

Reviewer 2 Report

This article aims at the problem of high-efficiency feature extraction of hyperspectral images, this paper proposes a two-stage hyperspectral analysis method based on total variation. This method is theoretically proved in the article. According to the experimental results, the comparison between the proposed method and existing methods shows a competitive performance of the proposed method. This is an interesting research paper. There are some suggestions for revision.

  1. The motivation is not clear. Please specify the importance of the proposed solution.
  2. The listed contributions are a little bit weak. Please highlight the innovations of the proposed solution.
  3. The pros and cons of existing solutions should be discussed. Additionally, more recently published solutions should be discussed, such as "Remote sensing image defogging networks based on dual self-attention boost residual octave convolution", Remote Sensing 13 (16), 3104, 2021;  "Real-World Image Denoising with Deep Boosting," in IEEE Transactions on Pattern Analysis and Machine Intelligence, vol. 42, no. 12, pp. 3071-3087, 1 Dec. 2020, doi: 10.1109/TPAMI.2019.2921548.   "Multi-Resolution Aitchison Geometry Image Denoising for Low-Light Photography," in IEEE Transactions on Image Processing, vol. 30, pp. 5724-5738, 2021, doi: 10.1109/TIP.2021.3087943.
  4. Please pay attention to the chart format. First, when a table encounters a cross-page problem, there is a specific way to deal with it, not like it is disconnected. Second, an image should not be too large that it fills the entire layout, and some images can be a little bit small and listed in a general map. All charts have the above two problems.
  5. The time complexity of the proposed algorithms should be discussed and compared with existing solutions.
  6. Is the "the anisotropic total variation model (ATVM)" on line 95 proposed for the first time, or is it a reference to an existing model? It is not mentioned in the previous article. Please add some specific explanations or appropriate quotations, otherwise it may make readers confused.
  7. Please specify how to obtain the suitable parameter values of the proposed solution.
  8. The description of "inner loop" in line 157 may not be omitted for brevity. Please add some appropriate or concise descriptions that can help readers better understand the composition and details of the algorithm structure.
  9. What is the experimental environment?
  10. The experimental results are not convincing. Please compare the proposed solution with more recently published methods to verify the innovations and performance advantages of the proposed method.

Reviewer 3 Report

This manuscript proposed a feature extraction method for hyperspectral image classification, which combined average fusion, ATVM, and ITVM, where the ATVM is extended from relative TV method, and ITVM was solved by Split Bregman algorithm. The performance of proposed method is good, and analysis of parameters and computation complexity is sufficient, however, there are some issues should be improved in the description of proposed method. Below are detail comments to authors:

  1. in section 2, the descriptions of ATVM and ITVM are confused. What is the dimension of F and R in equation (1) and (9)? And in line 145, how vF and vR are used to represent the vectors of F and R? Are they representing the pixel vector of input dataset?
  2. The purposes to utilize methodologies in each stage is not described sufficiently. What is your reason to use average fusion? And what is the role of ATVM and ITVM in proposed framework?

Reviewer 4 Report

The manuscript is well organized.

L 226-239: In the experimental study, the control of the parameters significantly influences the result, and a description of the default value of the parameters needs to be added.

In Table 3 and Figure 2, in the specific class of SVM, (e.g., Oats), and in the case of WMRF (e.g., Grapes_untrained) in Table 4 and Figure 3, the accuracy is very low, which affects the overall accuracy. If the result of the processing shows an abnormal class, it is necessary to check.

The processing results are presented in ‘Chapter 4. Parameter Analysis and Discussion’, but there is a lack of explanation and interpretation of Figures 5, 6, 7, 8. 9, 10, and 11. Quantitative visualization is also important, but it is also necessary to qualitatively describe the advantages of the proposed method.

Round 2

Reviewer 2 Report

All my concerns have been addressed. I recommend this paper for publication.